# Amino acid and viral binding by the high-affinity Cationic Amino acid Transporter 1 (CAT1) from *Mus musculus*

Mingda Ye [1], Zhu Liang [1,2], Daming Zhou[2,3,4], Ashley C. W. Pike [1], SiYi Wang[1], Dong Wang [1], Souvika Bakshi[1], Laurent Brooke [1], Eleanor P. Williams [1], Jonathan M. Elkins[1], Benedikt M. Kessler [1,2], David I. Stuart [2,3] & David B. Sauer [1] ✉

Arginine, lysine, and ornithine are critical to several fundamental aspects of organismal physiology, including protein structure and function, the urea cycle, and intracellular signaling. These cationic amino acids are imported by several membrane transporters, most notably the Cationic Amino acid Transporters (CATs) in the SLC7 family. Of these, CAT1 is also the receptor for two orthoretroviruses, and determines the host tropism for these viruses. Here, using a combination of CryoEM and in vitro biochemical techniques, we characterize the substrate recognition and transport of CAT1 from *Mus musculus*. Further, by determining the structures of MmCAT1 in complex with the receptor binding domain from the Friend Murine Leukemia Virus, we identify the key structural interactions that determine the virus' rodent-specific tropism.

Cationic amino acids are critical to protein structure and function, cellular physiology, and intracellular signaling. The respective amine and guanidino groups of the lysine and arginine side chains are fundamental to protein folding, and protein-protein and protein-nucleic acid interactions[1]. Metabolically, lysine is an essential amino acid in humans[2], while arginine and the non-proteinogenic amino acid ornithine are central to the Krebs-Henseleit cycle's conversion of ammonia to urea[3,4]. Arginine also regulates cell growth and metabolism through its potent activation of the mechanistic target of rapamycin complex 1 (mTORC1)[5]. In cell signaling, arginine is metabolized to produce nitric oxide, while ornithine is converted to polyamines, thereby modulating vasodilation[6], apoptosis[7], and neurotransmission[8].

Several proteins in mammalian cells import and export arginine, lysine, and ornithine, most notably members of the SLC7 gene family. These include the Cationic Amino acid Transporters 1–3 (CAT1–3), the Cationic and Neutral L-type amino acid transporters 1 and 2 (y⁺LAT1-2), and neutral and basic amino acid transporter 1 (b⁰,⁺AT1)[9]. The CAT transporters are sodium-independent, with CAT1 having the highest affinity, and the most sensitive to trans-stimulation, but with the slowest import rate[10]. The transporter is central to intracellular and paracrine signaling, co-localizing with nitric-oxide synthase in plasma membrane caveolae[11] and interacting with TM4SF5 during its arginine-dependent regulation of the mTORC1 complex[12]. Consistent with its essential physiological functions, knockout of the SLC7A1 gene in mice causes perinatal death with anemia and reduced body mass[13], while its overexpression is associated with ovarian and colorectal cancers[14,15]. In addition to their native role in amino acid metabolism, the orthologs of CAT1 are receptors for oncogenic retroviruses[16], including Friend murine leukemia virus (FrMLV) and other murine leukemia viruses[17,18], and the bovine leukemia virus[19]. CAT1 is the primary determinant of FrMLV tropism[20–22], with key sequence differences between orthologs explaining why the virus exclusively infects rodent species.

CAT1 belongs to the amino acid-polyamine-organocation (APC) superfamily of amino acid transporters[23]. Extensive studies of bacterial homologs have revealed a LeuT-like fold that mediates alternating access to a central binding site for amino acid transport[24–28].

[1]Centre for Medicines Discovery, Nuffield Department of Medicine, University of Oxford, Oxford, UK. [2]Chinese Academy of Medical Sciences Oxford Institute, Nuffield Department of Medicine, University of Oxford, Oxford, UK. [3]Division of Structural Biology, Centre for Human Genetics, University of Oxford, Oxford, UK. [4]Present address: College of Life Sciences, Zhejiang University, Hangzhou, China. ✉e-mail: david.sauer@cmd.ox.ac.uk

Understanding of amino acid transport by human SLC7 genes has been further illuminated by recent structures of b[0,+]AT1[29–31], xCT[-32,33], LAT1[34–36], and Asc1[37,38]. Structures of AdiC[39], engineered GkApcT[24], and b[0,+]AT1[29] have revealed these proteins' interactions with arginine, though the substrate's binding pose and coordination chemistry are quite distinct among the distant homologs. Furthermore, the CAT transporter genes contain a large C-terminal insertion encoding two additional transmembrane helices with unknown function relative to the other SLC7-family transporters[17], and their bacterial homologs[24].

Therefore, to resolve the sequence and structural differences of CAT transporters that underlie their distinct biochemical and functional differences, we determined MmCAT1's structure, apo and in complex with arginine, lysine, and ornithine. Further, viral attachment with solute carriers as receptors has only been described for 5 transporters[40–46]. To understand the high affinity and tropism of the FrMLV viral model system[47], we characterized MmCAT1's structural interactions with the virus's receptor binding domain (FrMLV-RBD).

## Results

### Overall structure of MmCAT1

To investigate the cationic amino acid transporters' viral binding and transport mechanisms, we expressed MmCAT1 and the previously crystallized FrMLV-RBD construct using standard methods (Supplementary Fig. 1a–d)[48,49]. Consistent with its substrate selectivity[17,50], MmCAT1 in detergent solution was stabilized by arginine, lysine, and ornithine, while cysteine and leucine had no effect (Fig. 1a and Supplementary Fig. 1e) and thermostabilization by arginine was concentration-dependent (Fig. 1b and Supplementary Fig. 1f). Notably, this melting curve has two inflection points at 50.9 and 81.6 °C, termed $T_M1$ and $T_M2$ respectively, with only the lower inflection point sensitive to the presence of substrate. This suggests the independent unfolding of two domains within MmCAT1, where only one domain binds the substrate. In vitro, MmCAT transports lysine in proteoliposomes, with the bacterial transporter LysP[51] as a positive control (Fig. 1c). The

purified MmCAT1 binds FrMLV-RBD with a dissociation constant of 9.7 nM (Fig. 1d), consistent with previous measurements[52]. The Alexa-555 labeled FrMLV receptor binding domain also colocalized with EGFP tagged MmCAT1 expressed on HEK293 (Fig. 1e–g), in agreement with previous studies of RBD binding[18]. This MmCAT1:FrMLV-RBD complex was stable on size exclusion chromatography (Supplementary Fig. 1g, h) and we therefore determined its structure in the presence of arginine to 2.8 Å by cryo-Electron Microscopy (cryo-EM) (Fig. 1h, i, and Supplementary Fig 2, Table 1).

The high-resolution cryo-EM map of the MmCAT1:FrMLV-RBD complex revealed a 1:1 complex and enabled straightforward model building for both proteins (Fig. 1h, i). We were able to build the entire transport protein except for residues 20–30 at the N-terminus, 431–465 in intracellular loop 5 (ICL5) between TM10 and H11a, and 605–622 at the C-terminus. Using the previously determined FrMLV-RBD structure[53] as a template, we were able to build residues 43–265 of the viral surface protein. Notably, being expressed in human cells, our FrMLV-RBD is glycosylated, and we were able to build N-linked glycans at Asn46[FrMLV-RBD] and Asn202[FrMLV-RBD]. Similarly, the transporter has glycans attached at Asn223[MmCAT1] and Asn229[MmCAT1].

MmCAT1 is composed of 14 transmembrane helices (Supplementary Fig. 3a, b). Of these, TMs 1–10 and 13–14 fold adopt the classic APC fold with a structure similar to GkApcT (backbone RMSD 1.5 Å)[24]. While all of the expected disulfide bonds were found in FrMLV-RBD, the inter-cysteine bond predicted by AlphaFold[54] between Cys226[MmCAT1] and Cys309[MmCAT1] is notably absent in our final reconstructed map (Supplementary Fig. 3c). However, the side chains are correctly oriented for this bond, and disulfides can be broken by radiation damage[55]. Therefore, we hypothesize this bond was present in the purified protein but broken during data collection.

MmCAT1 also contains the additional transmembrane helices TM11 and TM12 relative to the canonical APC fold, which appear unique to the CATs and the related GABA-transporting SLC7A14[9,56] (Supplementary Fig. 11). Within the MmCAT1 structure, these helices are

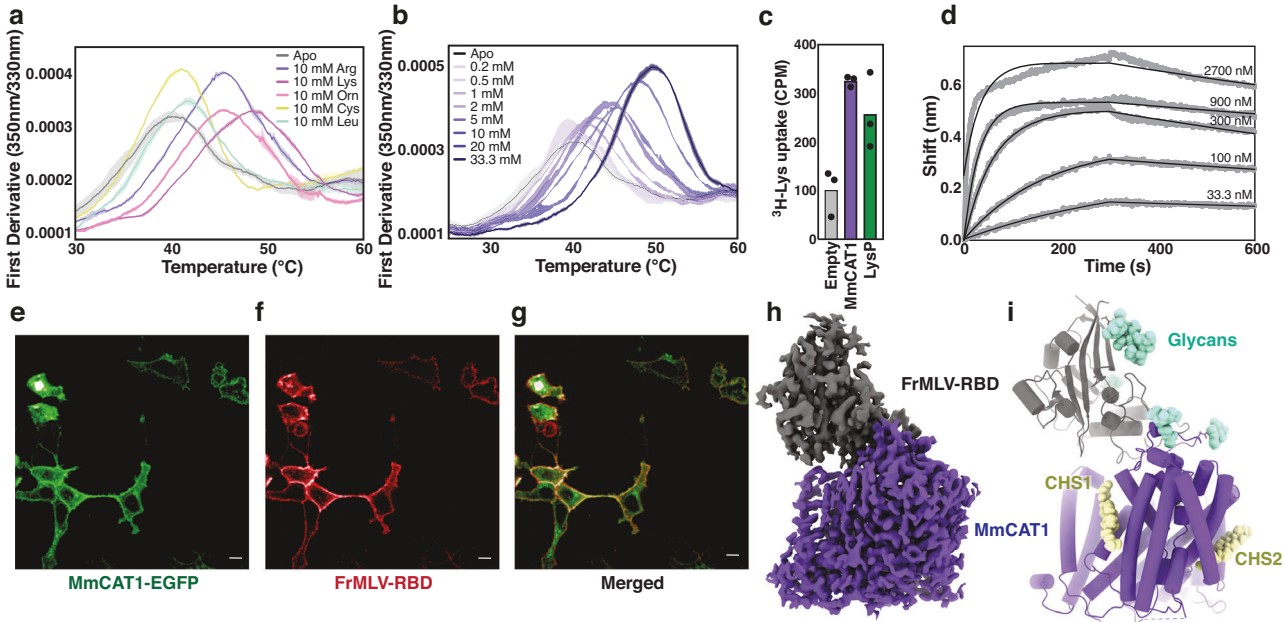

**Fig. 1 | MmCAT1 binds FrMLV-RBD and cationic amino acids in solution.** Binding of MmCAT1 to (**a**) various amino acids and (**b**) increasing concentrations of arginine measured by nano Differential Scanning Fluorescence (nanoDSF). Solid lines and shading in the same color correspond to the average and s.d. of three experiments. **c** Uptake of ³H-Lysine by MmCAT1 and LysP in proteoliposomes (N = 3). **d** FrMLV-RBD binding kinetics to MmCAT1 measured by BioLayer Interferometry. Confocal imaging of (**e**) EGFP-tagged MmCAT1, (**f**) Alexa555-labeled FrMLV-RBD, and (**g**) merged image. Representative image from 5 biological replicates. Scale bar is 10 μm. **h** Cryo-EM map and (**i**) experimental model of MmCAT1:FrMLV-RBD complex. Viral protein and transporter are colored gray and purple, respectively. The glycans and cholesterol hemisuccinate (CHS) are shown in cyan and yellow, respectively.

**Table 1 | Cryo-EM data collection, refinement and validation statistics**

| | MmCAT1:Arg:FrMLV-RBD (EMDB-50669) (PDB 9FQU) | MmCAT1:Lys:FrMLV-RBD (EMDB-50670) (PDB 9FQV) | MmCAT1:Orn:FrMLV-RBD (EMDB-50671) (PDB 9FQW) | MmCAT1(apo):FrMLV-RBD (EMDB-50668) (PDB 9FQT) |
|---|---|---|---|---|
| **Data collection and processing** | | | | |
| Magnification | 105k | 105k | 105k | 105k |
| Voltage (kV) | 300 | 300 | 300 | 300 |
| Electron exposure (e–/Å²) | 50 | 50 | 50 | 50 |
| Defocus range (μm) | -1.2 – -2.4 | -1.2 – -2.4 | -1.2 – -2.4 | -1.2 – -2.6 |
| Pixel size (Å) | 0.825 | 0.825 | 0.825 | 0.825 |
| Symmetry imposed | C1 | C1 | C1 | C1 |
| Initial particle picks (no.) | 2,375,463 | 3,150,614 | 1,567,413 | 2,482,628 |
| Particles after 2D (no.) | 772,292 | 684,030 | 579,080 | 517.673 |
| Final particles (no.) | 386,193 | 417,283 | 247,271 | 263,131 |
| Map resolution (Å) (GSFSC = 0.143) | 8.88 - 2.79 | 26.39 - 2.95 | 45.23 - 3.01 | 37.34 - 3.50 |
| **Refinement** | | | | |
| Initial model used | AlphaFold & 1AOL | AlphaFold & 1AOL | AlphaFold & 1AOL | AlphaFold & 1AOL |
| Model resolution (Å) (FSC = 0.5) | 3.0 | 3.1 | 3.2 | 3.8 |
| Map sharpening $B$ factor (Å²) | -120.3 | -138.7 | -129.2 | -155.1 |
| Model composition | | | | |
| Non-hydrogen atoms | 6059 | 6031 | 6031 | 5922 |
| Protein residues | 770 | 770 | 768 | 760 |
| Ligands | 13 | 11 | 12 | 11 |
| $B$ factors (Å²) | | | | |
| Protein | 61.64 | 62.96 | 66.79 | 66.80 |
| Ligand | 67.29 | 68.89 | 68.17 | 66.92 |
| R.M.S. deviations | | | | |
| Bond lengths (Å) | 0.002 | 0.002 | 0.002 | 0.003 |
| Bond angles (°) | 0.478 | 0.465 | 0.502 | 0.609 |
| Validation | | | | |
| MolProbity score | 1.24 | 1.27 | 1.23 | 1.49 |
| Clashscore | 4.63 | 5.15 | 4.57 | 9.17 |
| Poor rotamers (%) | 0.16 | 0.00 | 0.00 | 0.16 |
| Ramachandran plot | | | | |
| Favored (%) | 98.03 | 98.03 | 98.68 | 98.81 |
| Allowed (%) | 1.97 | 1.97 | 1.32 | 1.19 |
| Disallowed (%) | 0.00 | 0.00 | 0.00 | 0.00 |

peripheral to the core transport domain and coordinated nearly exclusively by van der Waals contacts with TM10 and TM13 (Supplementary Fig. 3d). However, the structure does not immediately suggest a functional role for TM11 and TM12, as they are distant from the attachment sites of MgtS to GkApcT[24], 4F2hc to LAT1, or rBAT to b[0,+]AT1 and, therefore, unlikely to play a similar role in transporter stability or activity. Further, these helices are distant from the standard APC cytoplasmic and extracellular gates, and therefore do not appear to regulate the transporter's enzymatic cycle. This is consistent with the bimodal melting curve of MmCAT1, suggesting $T_M1$ reflects the substrate-dependent unfolding of the APC domain while $T_M2$ corresponds to the separate, substrate-independent unfolding of TM11 and TM12.

We noticed two large additional densities attached to the transport domain of the MmCAT1. Based on their shape and location on the protein surface, we identified these as cholesterol hemisuccinate (CHS) from the purification buffer (Fig. 1i). The first (CHS1) interacts with transmembrane helices 2, 6a, and 13, and does not overlap with the cholesterol or cholesterol hemisuccinate binding sites in

structures of other SLC7-family transporters[57]. In contrast, the second (CHS2) interacts with TMs 4, 5a and 8b and is adjacent to the cholesterol binding site of LAT1[36], though in MmCAT1 this semi-synthetic sterol sits more deeply into the gap between TM4 and TM5.

## Mechanism of FrMLV high-affinity binding

The interface of MmCAT1 and the FrMLV receptor binding domain is dominated by extracellular loop 3 (ECL3) of the transporter and variable region A (VRA) of the viral protein, with additional contacts made by the RBD's VRA with ECL1 and H11c (Fig. 2a, Supplementary Fig. 3e). A surface area of 787 Å² is buried upon FrMLV-RBD binding, and ~16 hydrogen bonds are formed with the transporter. These are arranged roughly into three layers based on their contacts with MmCAT1. In the first layer, the transporter's TM1b and TM6a make three contacts with the viral protein's C helix (Fig. 2b). The second and innermost layer is the most extensive, with a number of hydrogen bonds formed by MmCAT1's residues between Asp230[MmCAT] and Glu237[MmCAT], in ECL3, and the viral protein between Gly116[FrMLV-RBD] and Asn137[FrMLV-RBD] (Fig. 2c). The final layer is formed by Glu221[MmCAT] and Lys222[MmCAT] on

ECL3, and immediately following H5b, with hydrogen bonds to Asn133[FrMLV-RBD] and Gln95[FrMLV-RBD] (Fig. 2d).

Notably, our transporter-bound FrMLV-RBD structure is nearly identical to the previous crystal structure of this domain (RMSD = 0.4 Å), with only modest movements that are directly or indirectly linked to MmCAT1 binding (Fig. 2e). Most notably, the structural motif corresponding to the VRA sequence motif undergoes a 4° rigid body rotation away from helix G upon receptor binding, with only the side chain of Arg119[FrMLV-RBD] moving significantly to make a salt bridge with Glu60[MmCAT1]. Similarly, the loop between strands 8 and 9 of the RBD moves to engage with the interfacial helix 11c of MmCAT1, while helices F and G also shift toward the transporter. This explains the high affinity of the viral protein for MmCAT1, providing a strong enthalpic driving force while the limited structural changes minimize entropic costs.

This structure explains the physico-chemical role of residues on MmCAT1 that have previously been noted as crucial to viral binding. Introducing MmCAT1's Val233[MmCAT1], Tyr235[MmCAT1] and Glu237[MmCAT1] into equivalent positions of human CAT1 is sufficient for that ortholog serve as a receptor for FrMLV[21]. The critical Tyr235[MmCAT1] makes hydrophobic contacts with Phe224[MmCAT1] and Val233[MmCAT1], while also hydrogen bonding with Gly116[FrMLV-RBD] through its side chain hydroxyl and Asp120[FrMLV-RBD] via its backbone amine. The side chain's intra-molecular contacts are likely most essential, as smaller residues cause $10^4$-$10^6$ fold changes in viral titer, while loss of the side chain hydroxyl results in only $10^1$-$10^2$ fold in infectivity[58]. The transporter's Glu237[MmCAT1] is next most impactful to MLV infection[21], and interacts with Thr129[MmCAT1], Asn137[MmCAT1], and Arg131[MmCAT1]. This position in MmCAT1 tolerates glutamate to aspartate mutation but not charge reversal[58], indicating that charge-charge repulsion at this site is sufficient to block viral binding. Finally, Val233[MmCAT1] makes short hydrogen bonds with the backbone of Cys117[FrMLV-RBD] and Arg119[FrMLV-RBD]. The human chimera with glycine at the equivalent position of Val233[MmCAT1] has significantly diminished viral infection[21], suggesting the side chain orients the peptide backbone to strengthen the viral-transporter interaction.

Examining the FrMLV receptor binding domain also reveals the role of its essential residues to interact with MmCAT1. The side chains of Ser118[FrMLV-RBD] and Asp120[FrMLV-RBD] interact directly, Asp120[FrMLV-RBD] hydrogen bond to the backbone amines of Tyr235[MmCAT1] and Gly236[MmCAT1], and orient Cys117[FrMLV-RBD] and Arg119[FrMLV-RBD] form backbone-backbone interactions with Val233[MmCAT1]. Accordingly, both Ser118[FrMLV-RBD] and Asp120[FrMLV-RBD] are critical to binding, with mutation abrogating RBD binding[59]. Similarly, the Trp136[FrMLV-RBD] side chain makes a hydrogen bond with the backbone carbonyl of Gly236[MmCAT1], and correspondingly, a mutation of this residue blocks the binding of the viral protein[59].

## Substrate binding and selectivity

Within the transport domain, we noticed additional density in the center of the MmCAT1 near the break in TM1 (Fig. 3a), in a region corresponding to the canonical binding site of the APC family. The shape of this density agreed well with the arginine added to the purification buffer, and we modeled it as the transporter's substrate (Fig. 3b).

Examining MmCAT1's coordination of arginine, the substrate's peptide amine is liganded by hydrogen bonds with the backbone carbonyls of Thr45[MmCAT1] on TM1 and Tyr257[MmCAT1], Ala258[MmCAT1], and Val260[MmCAT1] on TM6. The substrate's carboxyl group is also coordinated by hydrogen bonds to the backbone amines of Ala48[MmCAT1] and Gly49[MmCAT1], and carbonyl of Thr45[MmCAT1] on TM1, and a further additional hydrogen bond with the side chain of Ser343[MmCAT1] on TM8b. To validate the role of these interactions, we took advantage of the effect of substrate on $T_M1$ of MmCAT1 (Fig. 3c), where arginine increases the protein's melting temperature and apparent cooperativity of unfolding. Consistent with their role in coordinating the substrate's

carboxylate moiety, the S343V[MmCAT1] or Y257F[MmCAT1] mutations ablate the change in thermostability with arginine binding (Fig. 3c and Supplementary Fig. 4). Curiously, the mutants Y257K[MmCAT1] and Y257E[MmCAT1] retain arginine-induced thermostabilization, suggesting that substrate binding only requires the ability to hydrogen bond at this position.

Exploring the substrate's side chain coordination by MmCAT1, we noted the guanidino group hydrogen bonds with the hydroxyl and carbonyl groups of Ser120[MmCAT1] of TM3, and is nearby the side chain Asp263[MmCAT1] on TM6b (Fig. 3b). Confirming the proposed importance of this conserved, acidic moiety[24], the mutation D263K[MmCAT1] ablates arginine binding (Fig. 3c). In addition, we noticed a water molecule linked by hydrogen bonds to the substrate's guanidino group and the carbonyl of Ser44[MmCAT1] on TM1. Finally, several highly conserved residues line the side chain binding pocket. Of these, Val260[MmCAT1] and Met405[MmCAT1] are conserved in the CAT transporters (Supplementary Fig. 11) and make van der Waals contacts with the substrate (Fig. 3b). Confirming the importance of these residues in coordinating the substrate, the mutations V260I[MmCAT1], M405K[MmCAT1], and M405Y[MmCAT1] reduce or block arginine-induced thermostabilization of MmCAT1 (Fig. 3c). We also probed the effect of mutating Gly261[MmCAT1], which is immediately adjacent to the substrate's side chain. Introducing the larger phenylalanine side chain at this position blocks arginine thermostabilization, though absolute conservation of a glycine at this position within the APC family makes deciphering the exact function of this mutation difficult (Supplementary Fig. 11).

We next compared our structure to other arginine-bound APC-fold transporters. MmCAT1 has a similar mechanism for coordinating the amino acid carbonyl and amine as GkApcT (Supplementary Fig. 3f). More generally, the substrate-interacting residues are conserved in CAT1 homologs (Supplementary Fig. 3g and Supplementary Fig. 11). Nevertheless, the CAT1 and GkApcT appear to have a subtle difference in transport mechanism. In the bacterial protein, the terminal amine of Lys191's side chain occupies the classic Na2 site of the homologous LeuT (Supplementary Fig. 3h), and either lysine or asparagine at this position can bind substrate and drive transport[24]. However, substrate-dependent thermostabilization of MmCAT1 is ablated and significantly reduced for N197A[MmCAT1] and N197K[MmCAT1], respectively (Fig. 3c). This suggests arginine binding or MmCAT1's structural changes in response to substrate are defective in these mutants, and the mammalian CAT transporter transport substrates via a subtly different mechanism from GkApcT.

In order to further elucidate the substrate binding and selectivity mechanism of MmCAT1, we incubated the purified MmCAT1:FrMLV-RBD complex with lysine and ornithine, and determined their cryo-EM structures to 3.0 Å (Supplementary Fig. 7, 8, and Table 1). The lysine and ornithine bound states of MmCAT1 are highly similar to the arginine bound structure, with respective RMSDs at 0.4 Å and 0.3 Å (Fig. 3d). MmCAT1's binding sites are nearly identical when coordinating each cationic amino acid (Fig. 3e, f, and Supplementary Fig. 7e). The side chain amine groups of lysine and ornithine are coordinated directly by backbone carbonyls and/or side chains Ser44[MmCAT1], Ser343[MmCAT1], and Ser347[MmCAT1], instead of a water molecule in the arginine structure. The backbone amine and carbonyl groups of lysine and ornithine interact with the same residues of MmCAT1 as arginine, except that ornithine forms an additional hydrogen bond between its carbonyl and the side chain of Tyr257[MmCAT1]. Notably, ornithine has moved by 0.9 Å toward the cytoplasmic side of the binding site relative to arginine (Supplementary Fig. 7e). This possibly reflects a shift in lowest-energy binding pose for MmCAT1 when coordinating an amino acid with a shorter side chain. The subtle differences in coordinating the cationic amino acid substrates are supported by the differences in substrate-dependent thermostabilization of MmCAT1. Mutations of Asn197[MmCAT1], Tyr257[MmCAT1], and Gly261[MmCAT1] have similar changes in

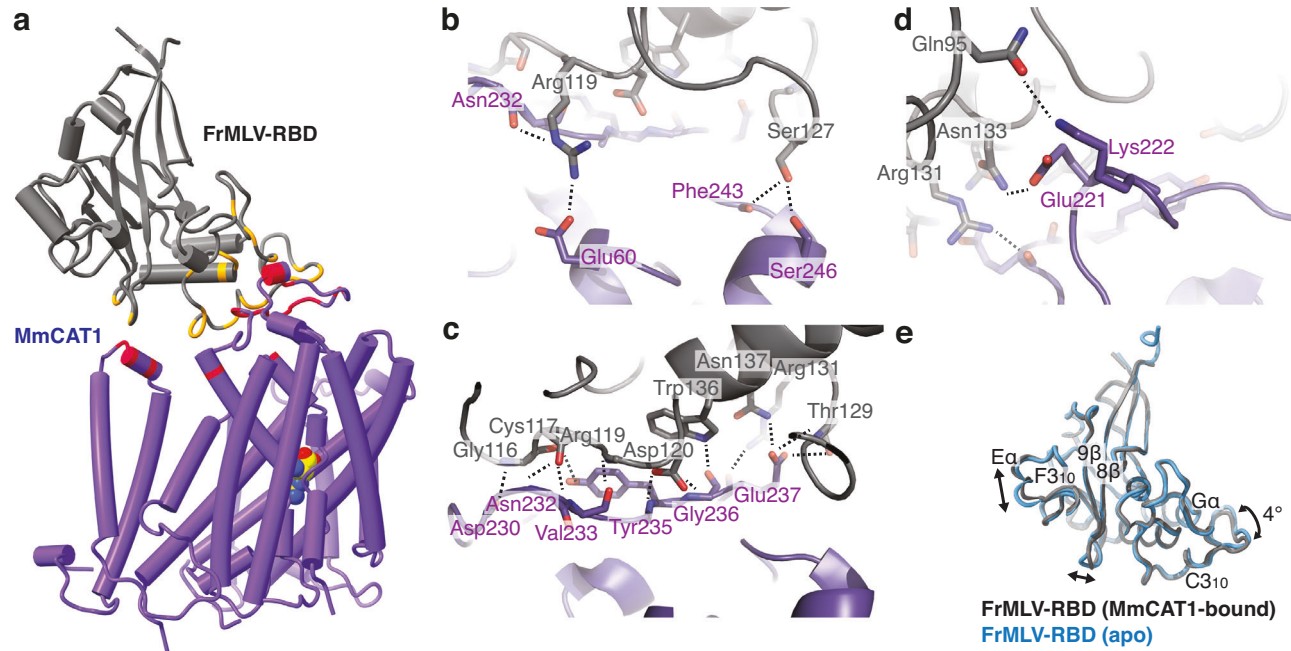

**Fig. 2 | FrMLV-RBD makes extensive hydrogen bonds with MmCAT1, particularly through ECL3. a** Structure of the MmCAT1:FrMLV-RBD complex with inter-protein contacts highlighted in red and gold, respectively. **b**–**d** Inter-protein hydrogen bonds between MmCAT1 and FrMLV-RBD. **e** Aligned structures of receptor-bound and apo FrMLV-RBD (PDB: 1AOL) colored in gray and blue, respectively.

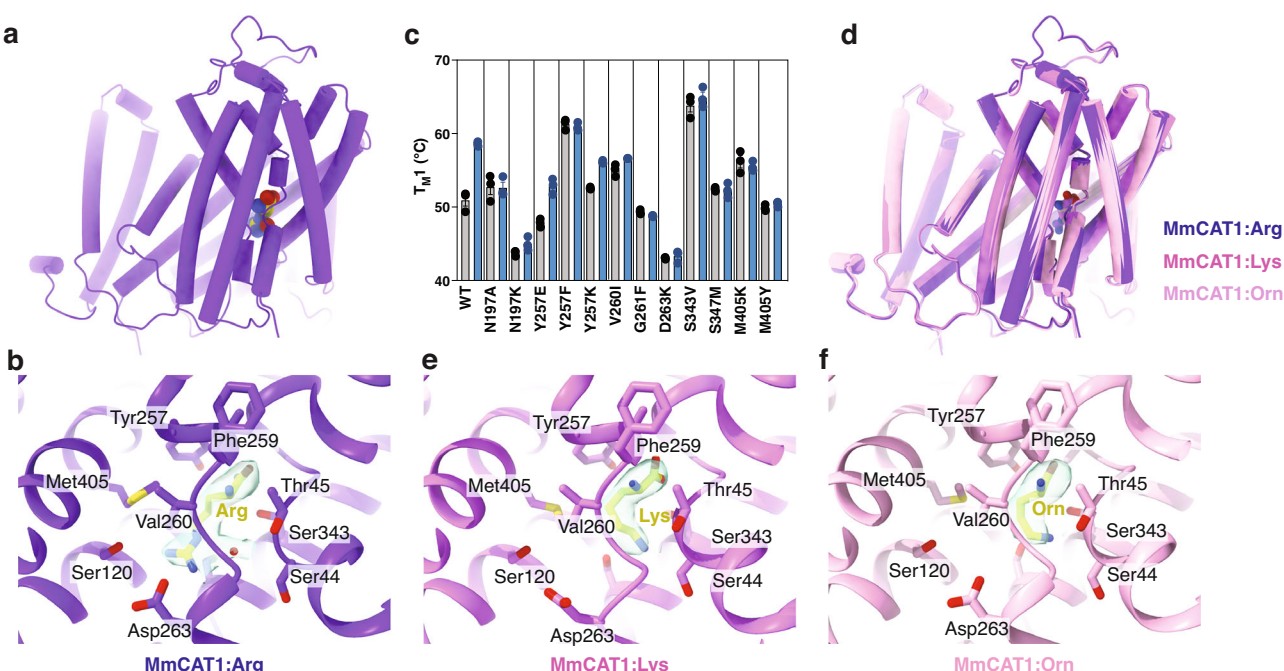

**Fig. 3 | MmCAT1 binds cationic amino acids in the active site of the transport domain. a** Structure of MmCAT1 bound to arginine, with protein and substrate shown as cartoon and spheres respectively. **b** Binding site of MmCAT1 purified with arginine. Substrate is shown as yellow sticks, while the substrate coordinating water is shown as a red sphere. Cryo-EM density for the substrate is shown as a semi-transparent surface. **c** First melting temperatures ($T_M1$) for MmCAT1 and mutants (N = 3). Melting temperatures with and without substrate are shown as blue and gray bars, respectively. **d** Overlay of MmCAT1 structures determined with arginine, lysine, and ornithine. Binding sites of MmCAT1 purified with (**e**) lysine and (**f**) ornithine.

thermostability upon addition of arginine or lysine (Supplementary Figs. 4, and 5), reflecting the interaction of these residues with shared moieties of both amino acids. In contrast, V260I$^{MmCAT1}$ and M405Y$^{MmCAT1}$ both have more pronounced increases in thermostability in response to lysine than arginine. This corresponds to each native residue's closer interaction with arginine side chain than with the substrate lysine.

## Conformational changes upon substrate binding/release

In our MmCAT1:FrMLV-RBD structure bound to arginine, the binding site is inaccessible from either the cytoplasm or extracellular space (Fig. 4a) in an inward-facing occluded state. To further probe the CAT1 transport cycle, we purified the MmCAT1:FrMLV-RBD complex without amino acid in the purification buffer and determined its structure to a resolution of 3.5 Å (Supplementary Fig. 8, and Table 1).

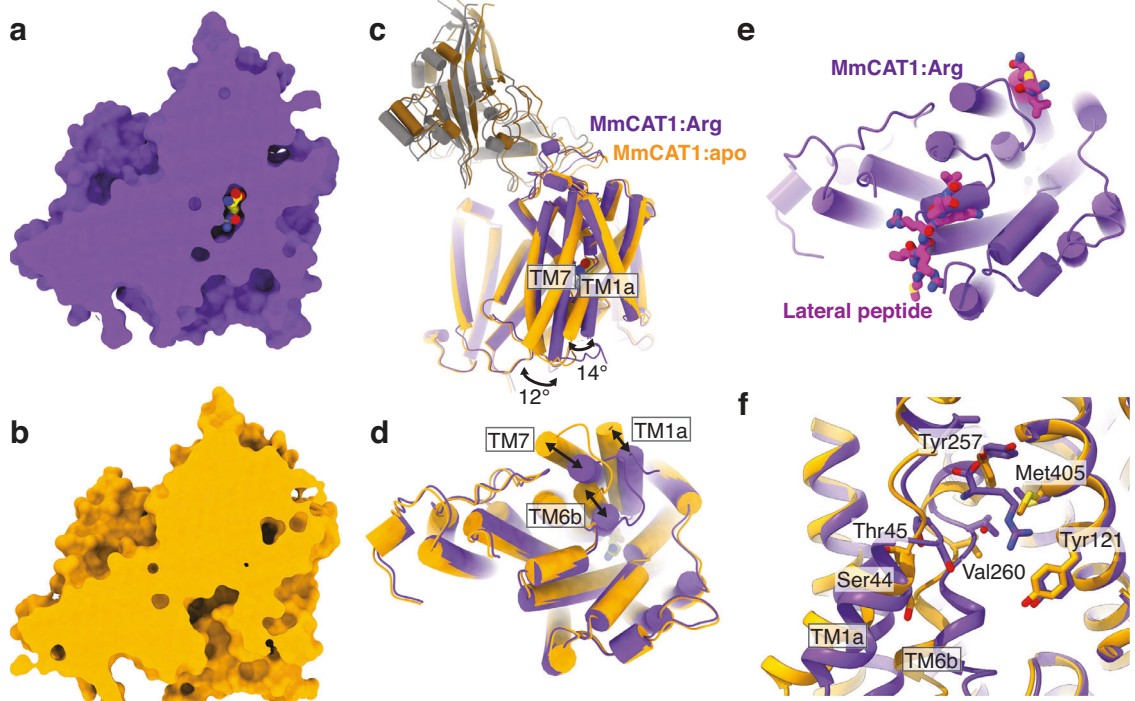

**Fig. 4 | Structural changes between MmCAT1's inward-occluded and inward-open conformations.** Structure of MmCAT1 determined in the presence of (**a**) arginine and (**b**) absence of substrate. Structural alignment of MmCAT1 in arginine-bound and apo states as viewed from (**c**) the membrane plane and (**d**) the cytoplasmic surface. **e** Structure of the lateral peptide over the cytoplasmic surface of arginine-bound MmCAT1. **f** Detailed view of the substrate binding site for arginine-bound and apo MmCAT1, shown in purple and yellow, respectively.

Comparing arginine-bound and substrate-free structures, we noted this in the absence of arginine the binding site is open to the cytoplasm, thereby revealing an inward-facing conformation of MmCAT1 (Fig. 4b). Opening the cytoplasmic gate is driven by large movements of TM1a, TM6b, and TM7 (Fig. 4c, d). TM1a has undergone a 14° rigid body swing away from TM5b, pivoting approximately at the TM1a-TM1b break. TM6b similarly pivots at the TM6a-TM6b break, swinging by 22° away from TM8b. Finally, pushed by the movement of TM1a and TM6b, TM7 rotates by 12° about Gly296[MmCAT1].

We also noticed changes on the extracellular side of the protein which appear coupled to the intracellular gate (Supplementary Fig. 9a, b) and thereby distinct from the proposed transport cycles for GkApcT and AdiC[24,25]. In MmCAT1's inward-facing occluded state, the extracellular gate is composed of TM1's Leu53[MmCAT1] and TM6a's Tyr257[MmCAT1], and TM10's Leu398[MmCAT1] (Supplementary Fig. 9a). The inward-occluded to inward-open conformational change leads to subtle changes of these residues, including Tyr257[MmCAT1] which breaks a hydrogen bond with Ser127[MmCAT1] and moves into the substrate binding site. The changes also lead to a rigid body movement of TM1b, extracellular loop 3, and the virus' receptor binding domain (Fig. 4c and Supplementary Fig. 9a). Though the function of this movement in MmCAT1's transport cycle is unclear, the interface between transporter and FrMLV-RBD appears preserved, suggesting that transporter binding to the receptor binding domain and substrate are independent.

While opening MmCAT1's cytoplasmic gate involves large movements of TM1a, TM6b, and TM7, a notably absent structural motif in either state is the lateral helix identified in GkApcT (Supplementary Fig. 9c, d)[24]. While our construct was designed to include this helix (Supplementary Fig. 11), instead we found this region forms a short loop in the substrate-bound states interacting with the cytoplasmic faces of H3a, TM3b, TM8c, and TM14 (Fig. 4e, Supplementary Fig. 9c, e). In this substrate-free state, the same region lacks any apparent

density (Supplementary Fig. 9f), suggesting it becomes disordered after being pulled away from the hash domain of MmCAT1's APC fold by the movement of TM1a. Therefore, we hypothesize this region of the protein, termed the lateral peptide, reversibly latches the CAT transporter's TM1a gating helices to the core hash domain[28], and thereby regulating the inward-occluded to inward-open conformational change. Supporting this notion, the transport activity of MmCAT1 is blocked by the mutation of Glu107[MmCAT1] at the cytoplasmic face of TM3b and immediately adjacent to the lateral peptide[60]. Similarly, TM8c was noted as critical to the affinity for arginine of HsCAT1 and HsCAT2 chimeras[61], although in our homologous structure the two critical residues are oriented away from the lateral peptide and therefore may act indirectly.

While the overall structure of the complex is similar, the empty MmCAT1 binding site indicates we have captured an apo state of the transporter (Supplementary Fig. 9g, h). The movement of TM1 and TM6b leads to the displacement of hydrogen bonding groups from those helices by an average of 4 Å and 2 Å, respectively (Fig. 4f). Disrupting these six hydrogen bonds would dramatically decrease the affinity of the inward-facing state for arginine and thereby drive substrate release. Furthermore, these broken hydrogen bonds are exclusively with the arginine's backbone amine and carboxylate. This immediately suggests that inward-occluded to inward-open conformational change should be insensitive to the substrate side chain, in agreement with the similar transport rates for arginine, lysine, and ornithine[50].

## Discussion

Here, we determine structures of a CAT1 protein and identify residues in the binding site that are key to the transporter's selectivity to arginine, lysine, and ornithine. These substrate-interacting residues were validated with targeted mutagenesis, though our binding assays cannot determine if the binding-competent mutants retain transport

activity. Furthermore, by also capturing a substrate-free inward-open structure, we identify a key sequence at the N-terminus of CAT proteins that dynamically interacts with the intracellular face of the protein and thereby likely regulates transporter activity. Finally, by capturing the structure of MmCAT1 in complex with FrMLV-RBD, we note that the viral protein's nanomolar affinity is enthalpically driven by numerous hydrogen bonds while its rigid structure minimizes entropic costs.

FrMLV's receptor binding domain engagement with MmCAT1 also explains the variable host tropism and receptors for the Gammaretrovirus genus. The RBD's primary interaction with MmCAT1's ECL3 is quite distinct from the viral attachment of NPC1[62], B[0]AT1-ACE2 and SIT1-ACE2 complexes[40,41], NTPC[44], MFSD2A[45], and ASCT2[46]. In contrast to the attachment of these viral proteins to structured and conserved domains, FrMLV-RBD binds primarily through a stretch of ECL3 between 230 and 237 which is poorly conserved across homologs (Supplementary Fig. 10a). Highlighting the importance of this region to RBD binding, and consistent with previous viral infection and envelope protein binding studies[20,21], we found that fluorescently labeled FrMLV-RBD only bound to human or mouse CAT1 chimeras containing the mouse ECL3 sequence between 227 and 237 (Supplementary Fig. 10b–d). The distant binding site of FrMLV-RBD from the transport domain, and poor conservation of this region, indicate that MmCAT1's viral binding and amino acid import activities are independent. This agrees with previous studies showing MmCAT1's amino acid transport is not affected by binding to the virus or its envelope glycoprotein[50], and mutations in the APC domain that block amino acid transport do not affect the glycoprotein binding[60]. In contrast, Bovine Leukemia Virus (BLV) can use several mammalian CAT1 orthologs as receptors[19] and blocks amino acid import[63]. This is consistent with the distant relationship of BLV and FrMLV envelope proteins (Supplementary Fig. 10e), and we hypothesize the BLV deltaretrovirus RBD attaches to CAT1 through a more conserved region of the transport domain. Furthermore, we noted RBD helices E and F are coupled to the loop between strands 8 and 9 by a hydrogen bond of Tyr153[FrMLV-RBD] and Asp251[FrMLV-RBD] (Supplementary Fig. 10f). This structural link indirectly couples helical movements to receptor binding, and suggests the corresponding VRC sequence motif may modulate the viral RBD's binding affinity and kinetics despite not directly contacting MmCAT1.

The structure of MmCAT1 provides a hint for the cationic amino acid selectivity of CAT transporters, which do not transport the neutral glutamine or unprotonated histidine[17]. Asp263[MmCAT1] is the only charged residue of the binding site and is essential for arginine binding by MmCAT1. Furthermore, this residue is conserved in CATs 1-4, SLC7A14, and the arginine transporting y[+]LAT1, y[+]LAT2, StAdiC, EcAdiC, and GkApcT. Therefore, despite not directly interacting with the substrates' side chains in our inward-occluded structures, we hypothesize electrostatic interactions with this residue will significantly increase the protein's selectivity for amino acids with cationic side chains. Notably, b[0,+]AT1 has an asparagine at the equivalent position but also an aspartate at the equivalent of Val260[MmCAT1] (Supplementary Fig. 10g), suggesting it uses the same electrostatic mechanism to bind cationic substrates. Further, we noted the Ser343[MmCAT1] coordinates the water in the binding site, while b[0,+]AT1's has an alanine at the equivalent position and the binding site is dehydrated. This serine, absolutely conserved in CAT1-3, explains their selective exclusion of neutral amino acids via a more polar and hydrated binding site. Our biochemical results suggest a similar role for Ser347[MmCAT1], which is essential for arginine-dependent thermostabilization of MmCAT1 (Supplementary Fig. 4) despite its hydroxyl not directly coordinating the cationic substrate in the inward-occluded structures. As it is nearby the arginine's guanidino group and conserved in the CAT1-3 proteins, we hypothesize it may transiently interact with substrate during loading and release. The binding poses of the substrate cationic amino acids to MmCAT1 also explains why the transporter is selective for

L-arginine and L-lysine over their D stereoisomers[17,50]. While MmCAT1's binding site can accommodate a D amino acid's backbone amine and carboxylate, its side chain directly clashes with the nearby TM1 and TM8.

Notably, N-Ethylmelamide (NEM) is a CAT-specific inhibitor of amino acid import and known to modify the equivalent of Cys31[MmCAT1] and Cys264[MmCAT1] [64]. Our structures reveal that these residues are adjacent to or within the gating helices TM1a and TM6b (Supplementary Fig. 10h). Therefore, chemical modification of either residue may block access to the binding site or closing of the cytoplasmic gate, thereby arresting the transport cycle.

Finally, our results suggest a mechanism for CAT1's role in cellular sensing of amino acid depletion, where MmCAT1's large arginine-dependent conformational changes correspond to its substrate-dependent interactions with TM4SF5 to signal the cells' arginine status to the mTOR complex[12]. This is consistent with the transceptor function reported for insect CAT homologs[65,66], though the interactions of mammalian CAT1, TM4SF5, and arginine with other components of the mTOR complex to signal cationic amino acid availability will require further study.

During the final preparation of this manuscript, a complementary manuscript was published and interested readers can find it here (https://doi.org/10.1038/s41467-025-67704-6).

## Methods

### Sequence alignment and analysis
Sequences of SLC7 family proteins and their prokaryotic prototypes were aligned in PROMALS3D[67]. The same workflow was used to align the full-length envelope protein sequences from select deltaretroviruses, gammaretroviruses, and endogenous retroviruses, the phylogenetic tree calculated using FastTree 2[68], rooted using the envelope protein of Rous sarcoma virus.

### SLC7A1 cloning
The full-length SLC7A1 gene from *Mus musculus* was synthesized after codon optimization for *Homo sapiens*, and full-length (Met1-Lys622) and N-terminally truncated (Met13-Lys622) constructs were subcloned into an in-house developed BacMam expression vector pHTBV1.1-CTGFP-SIII-10H which fuses a tobacco etch virus (TEV) protease cleavage site followed by EGFP, twin-strep and 10X His tags. Cloning of the full-length human SLC7A1 (Met1-Lys629) into pHTBV1.1-C3CGFP-SIII-10H-GTW, which fuses a 3C protease cleavage site followed by EGFP, twin-strep and 10X His tags, and all baculovirus production followed standard methods[48].

### MmCAT1 expression and purification
The resulting baculovirus was used to infect Expi293F cells in Freestyle 293 expression medium (GIBCO) in the presence of 5 mM sodium butyrate. Infected cells were grown in an orbital shaker for 48 h at 37 °C, 8% $CO_2$ and 75% humidity, harvested by centrifugation (1500x g for 15 min), washed with phosphate-buffered saline, flash-frozen, and stored at -80 °C until further use.

The cell pellets were resuspended in extraction buffer (50 mM HEPES, pH 7.5, 300 mM NaCl, 5% Glycerol) in the presence of cOmplete Protease Inhibitor Cocktail tablets (Roche). The pellet suspension was homogenized before adding LMNG and CHS to final concentrations of 1% and 0.2% respectively, and solubilized at 4 °C for 1–2 h with gentle rotation. The insoluble materials were pelleted at 55,000 × $g$ for 30 min. The supernatants were incubated with pre-equilibrated StrepTactin Superflow resin (IBA-Lifesciences) for 2 h to allow thorough binding of MmCAT1.The resin was then poured into a gravity column and washed with column buffer (50 mM HEPES, pH 7.5, 300 mM NaCl, 5% Glycerol, 0.003% LMNG, 0.0006% CHS). Protein was eluted with column buffer supplemented with 25–50 mM desthiobiotin (IBA-Lifesciences). Eluted MmCAT1 was either flash-frozen or

immediately concentrated using a centrifugal concentrator with 100 kDa cut off (Sartorius) and subjected to size exclusion chromatography using a Superdex 200 Increase 10/300 GL column (Cytiva) pre-equilibrated with MmCAT1-SEC buffer (10 mM HEPES, pH 7.5, 150 mM NaCl, 0.003% LMNG, 0.0006% CHS). Peak fractions corresponding to MmCAT1 were pooled and flash-frozen with liquid nitrogen until further use.

## LysP expression and purification

The LysP protein from *Pseudomonas aeruginosa* was purified as previously described[51], with the following modifications. Cell were disrupted by sonication on ice, the TEV digestion and dialysis steps omitted to retain the GFP tag, and the protein was purified on size exclusion chromatography on Superdex 200 pre-equilibrated in a buffer containing 20 mM HEPES (pH 7.5), 150 mM NaCl, and 0.025% (w/v) DDM.

## In vitro ³H-Lys uptake by MmCAT1

To prepare liposomes, a total of 100 mg of *E. coli* polar lipids (Avanti Polar Lipids) was first dissolved in 4 mL of chloroform in a round-bottom glass flask, and the chloroform was evaporated using Rotavap in a 40 °C water bath for around 30 min. The glass flask was put in a vacuum at room temperature overnight to evaporate residual chloroform. The next day, 3 mL of reconstitution buffer (50 mM phosphate pH 7.6, 1 mM DTT) was added to the glass flask, and the suspension cyclically freeze-thawed 10 times using liquid nitrogen and a 60 °C water bath. The liposome suspension was then extruded through a polycarbonate filter with 400 nm pores 23 times. This unilamellar liposome stock was snap frozen and stored at -70 °C until further use.

For the proteoliposome reconstitution, 300 μL empty liposome stocks were mixed with 0.06 mg LysP or full-length MmCAT1, both GFP tagged, or reconstitution buffer and supplemented with 1.25% n-octyl-β-D-glucoside (Anatrace). The liposome-protein-detergent mixture was incubated on ice for 45 min before being diluted into 35 mL of reconstitution buffer. The proteoliposomes were recovered by centrifugation at 170,000 × *g* for 1 h at 4 °C, resuspended in 100 μl reconstitution buffer, and used fresh for uptake experiments.

To initiate the transport assay, 2.66 μL of ³H-Lys (American Radiolabeled Chemicals) was added to 300 μL uptake buffer (50 mM potassium chloride pH 7.6, 1 mM DTT), followed by the addition of 25 μL proteoliposomes. After 50 min at room temperature, proteoliposomes from 95 μL of the reaction mix were collected on 0.22 μm MCE filter membranes (Merck) by filtration, washed with ice-cold reconstitution buffer, and radioactivity quantified on a Microbeta liquid scintillation counter.

## FrMLV-RBD protein production

The *Homo sapiens* codon-optimized gene of the FrMLV envelope protein was synthesized and the FrMLV-RBD construct (Ala35-Pro270) was subcloned into pHR-CMV-TetO2_His6_IRES-EmGFP. Protein was expressed in adherent HEK293T cells using an in-house developed lentiviral expression method[49]. Media with secreted FrMLV-RBD proteins was dialyzed against MLV buffer (50 mM HEPES, pH 6.9, 300 mM NaCl, 5% Glycerol) and concentrated to 200 mL and supplemented with 30 mM imidazole for subsequent purification. The concentrated supernatant was incubated with Ni-NTA resin pre-equilibrated with MLV buffer supplemented with 30 mM imidazole for 2 h at room temperature. Resin was washed thoroughly on a gravity flow column using MLV buffer supplemented with 20 mM imidazole before elution with MLV buffer supplemented with 500 mM imidazole. Eluted FrMLV-RBD was concentrated using a centrifugal concentrator with 10 kDa cut off (Sartorius) and subjected to size exclusion chromatography using a Superdex 200 Increase 10/300 GL column (Cytiva) pre-equilibrated with RBD-SEC buffer (10 mM HEPES, pH 7.5, 150 mM

NaCl). Peak fractions corresponding to FrMLV-RBD were pooled and flash-frozen with liquid nitrogen until further use.

## MmCAT1:FrMLV-RBD complex formation

Affinity-purified MmCAT1 was mixed with SEC-purified FrMLV-RBD at a molar ratio of 1:2, concentrated using a centrifugal concentrator with 100 kDa cut off (Sartorius), and subjected to size exclusion chromatography using a Superdex 200 Increase 10/300 GL column (Cytiva) pre-equilibrated with SEC buffer supplemented with 0.003% LMNG and 0.0006% CHS. Peak fractions corresponding to the MmCAT1:FrMLV-RBD complex were pooled and concentrated for subsequent experiments.

## Structure determination

The MmCAT1:FrMLV-RBD arginine, lysine, and ornithine complexes were obtained by supplementing the purified apo complex with 34 mM arginine, 20 mM lysine, and 20 mM ornithine, respectively, for 30 min on ice prior to grid freezing. Those samples at 6.36 mg/ml were applied to Quantifoil Gold 1.2/1.3, 300 mesh grids. The grids were blotted using a Mark IV Vitrobot (Thermo Fisher Scientific) at 4 °C and 100 % humidity with force of -10, 5-sec waiting time, and 7 to 9-second blot times, followed by plunging into liquid ethane.

Cryo-EM grids of the MmCAT1:FrMLV-RBD apo complex was prepared by applying the freshly pooled SEC-purified complex to a Quantifoil Copper 1.2/1.3, 200 mesh grid at 3.8 mg/ml. The grids were blotted using a Mark IV Vitrobot (Thermo Fisher Scientific) at 4 °C and 100 % humidity with force of -10, 5-sec waiting time, and 7 sec blot time, followed by plunging into liquid ethane.

Cryo-EM data were collected using a Titan Krios (Thermo Fisher Scientific) operating at 300 kV using a GIF-Quantum energy filter with a 20 eV slit width (Gatan) and a K3 direct electron detector (Gatan). All data were automatically collected using EPU software (Thermo Fisher) with target defocus ranges of -1.2 μM to -2.6 μM for the apo complexes, and -1.2 μM to -2.4 μM for the arginine, lysine and ornithine complexes. Each micrograph was dose-fractioned into 50 frames, with an accumulated dose of 50 e⁻/Å².

CryoSPARC 3.3.1 was used for all data processing workflows[69]. Movies were patch motion corrected, and CTF-corrected and manually curated based on ice thickness and CTF fit resolution. Particles were blob-picked, followed by 2D classification. The particles from well-resolved 2D classes were used for template-based picking followed by further 2D classification. The best-resolved particles were then used as training sets for several iterations of Topaz picking[70]. The well-resolved classes were used for ab initio reconstruction followed by heterogeneous refinement and non-uniform refinement. Particles from the first good non-uniform refinement results were used as training sets for further Topaz picking, followed by 4–5 rounds of heterogeneous refinement to filter out bad images and non-uniform refinements. Finally, local refinement was performed using masks around the proteins to minimize the errors caused by detergent micelles. For the apo structure, further global CTF refinement was performed. Initial models for MmCAT1 and the FrMLV-RBD came from AlphaFold[54] and the deposited crystal structure (PDB: 1AOL)[53], respectively. The models were rebuilt as necessary in COOT[71] and then refined with PHENIX real-space refinement[72].

To compare the densities of the MmCAT1 latch peptide and within the substrate binding site, maps were scaled to equivalent contours using the FrMLV-RBD as an internal reference, as previously described[73]. Sequence conservation mapped onto the proteins' structures was calculated using ConSurf with default settings[74,75].

## DSF binding assay

For DSF binding assays, mutants were generated on the full-length MmCAT1 construct in pHTBV1.1-CTGFP-SIII-10H. Wild-type and mutant proteins were then expressed and purified as described above in a

buffer of 10 mM HEPES pH 7.5, 150 mM NaCl, 0.003% LMNG, and 0.0006% CHS.

All proteins were incubated in test conditions for 30 min at room temperature prior to analysis. Amino acid stabilization screening was carried out using 1 μM wild-type truncated MmCAT1 with or without 10 mM arginine, lysine, ornithine, leucine, and cysteine. For the arginine titration experiment, 1 μM truncated MmCAT1 was incubated with a gradient of 0.2–33.3 mM arginine. For the substrate-binding site variants, 1 μM wild-type and mutant MmCAT1 proteins were incubated with or without 34 mM arginine or lysine. After incubation, samples were loaded into standard grade nanoDSF capillaries (Nanotemper), measured in Prometheus NT.48 device (Nanotemper) with excitation power 100 % and temperature gradient from 20 °C to 95 °C with a slope of 1 °C/min. Data were analyzed using PR ThermControl software (Nanotemper). All measurements were performed in technical triplicates.

### In vivo binding assay
FrMLV-RBD was labeled with Alexa555 by incubating SEC-purified FrMLV-RBD with NHS-Alexa555 (Lumiprobe) at the molar ratio of 1:1 for 2 h at room temperature, followed by overnight dialysis against MLV-SEC buffer at 4 °C.

ECL3 chimeras of CAT1 were generated on the truncated MmCAT1 and full-length HsCAT1 constructs in pHTBV1.1-CTGFP-SIII-10H and pHTBV1.1-C3CGFP-SIII-10H-GTW, respectively, were generated using site-directed mutagenesis. ECL3 sequences for the chimeras are listed in Supplementary Table 1.

HEK293T cells were seeded in poly-D-lysine coated 96-well plates (PhenoPlate microplates, Revvity), followed by transient transfection with MmCAT1 constructs (0.1 μg/well) using Lipofectamine 3000 (Thermo Fisher Scientific) according to manufacturer's instructions. After 48 h of transfection, medium was decanted, and cells washed three times with PBS. Cells were subsequently incubated with FBS-free medium containing 0.015 μM FrMLV-RBD-AF555 at room temperature for 1 h, washed three times with PBS, and imaged on a high-content laser-based spinning disk confocal microscope (Opera Phenix Plus, Revvity) using a 60x water objective. Images were processed and analysed using Harmony software (v5.9) following the analysis pipeline in Supplementary Table 2. 100 images were taken per well and data was collected from three replicates for each condition. Normalized co-localization intensity was calculated as the ratio of the mean of fluorescence emission intensities at 555 nm and 509 nm.

### FrMLV-RBD binding by BLI
Purified FrMLV-RBD was dialyzed in the BLI buffer (25 mM HEPES, pH 7.5, 125 mM NaCl, 0.003% LMNG, 0.0006% CHS) overnight. SEC-purified MmCAT1 containing the twin-strep and GFP tag was diluted to 100 nM in the BLI buffer before use. A 3-fold serial dilution of FrMLV-RBD with BLI buffer was used to make BLI samples with viral protein concentrations of 2.7 μM, 900 nM, 300 nM, 100 nM, and 33.3 nM. An additional, FrMLV-RBD free sample was prepared for background subtraction.

Bio-layer Interometry was then performed using the Gator Pivot (Gator Bio). Streptavidin-coated biosensors Strep-Tactin XT probes (GatorBio) were first incubated in BLI buffer for 1 min, then dipped into 100 nM MmCAT1 solution for 2 min, and finally rinsed with BLI buffer again for 1 min. Association was then measured by dipping the array of sensors into the six FrMLV-RBD BLI samples, including background control, for 5 min. Finally, dissociation was measured for 5 min by transferring the sensor array into protein-free BLI buffer. All incubation steps were performed at room temperature with three technical replicates.

FrMLV-RBD binding kinetics and dissociation constants were calculated by non-linear regression using Graphpad Prism 10.2.3 after background subtraction and aligning all traces to the beginning of the association step.

### Reporting summary
Further information on research design is available in the Nature Portfolio Reporting Summary linked to this article.

## Data availability
The data that support this study are available from the corresponding authors upon request. The cryo-EM maps and models generated in this study have been deposited in the EMDB database and the Protein Data Bank, respectively. The cryo-EM maps have been deposited in the Electron Microscopy Data Bank (EMDB) under accession codes EMDB-50669 (MmCAT1:Arg:FrMLV-RBD), EMDB-50670 (MmCAT1:Lys:FrMLV-RBD), EMDB-50671 (MmCAT1:Orn:FrMLV-RBD), EMDB-50668 (MmCAT1(apo):FrMLV-RBD). The atomic coordinates have been deposited in the Protein Data Bank (PDB) under accession codes PDB-9FQU (MmCAT1:Arg:FrMLV-RBD), PDB-9FQV (MmCAT1:Lys:FrMLV-RBD), PDB-9FQW (MmCAT1:Orn:FrMLV-RBD), PDB-9FQT (MmCAT1(apo):FrMLV-RBD).

Previously published structures for 1AOL (MLV-RBD), 6F34 (GkApcT), 6LI9 (b$^{0,+}$AT1:rBAT), 6IRT (LAT1:4F2hc), and 6XWM (LeuT). A source data file is available with this manuscript. Source data are provided with this paper.

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

## Acknowledgements

We are grateful to Simon Newstead for critical feedback on the manuscript. We would like to thank Val Millar and Daniel Ebner for access and help with the Perkin Elmer Opera Phenix instrument, and Malkeet Sahota, Denise Rohlik, and Benjamin Osborn of Gator Bio for access and support with the GatorPivot instrument and data collection. We thank Gamma Chi for assistance with CryoEM data processing. We are grateful to Emmanuel Nji for the LysP plasmid, expression protocol, and advice for the ³H-Lysine transport assay. This project has received funding from the Innovative Medicines Initiative 2 Joint Undertaking (JU) under grant agreement No 807015. The contents of this publication represent the views of the authors, and the JU is not responsible for any use that may be made of the information it contains. The JU receives support from the European Union's Horizon 2020 Research and Innovation Program and EFPIA. A.C.W.P. and J.M.E., and D.B.S. were supported by the Innovative Medicines Initiative 2 Joint Undertaking under grant agreement No 875510. The JU receives support from the European Union's Horizon 2020 research and innovation program and EFPIA and Ontario Institute for Cancer Research, Royal Institution for the Advancement of Learning McGill University, Kungliga Tekniska Hoegskolan, Diamond Light Source Limited. ZL, BMK and DIS are supported by the Chinese Academy of Medical Sciences (CAMS) Innovation Fund for Medical Science (CIFMS), China (grant number: 2024-I2M-2-001-1). Electron microscopy was provided through the Oxford Particle Imaging Center (OPIC), an Instruct-ERIC center (funded by Wellcome Trust JIF award [060208/Z/00/Z] and equipment grant [093305/Z/10/Z]) and the Electron Bio-Imaging Center, Diamond Light Source Ltd (eBIC; BAG proposal bi28713).

## Author contributions

M.Y., D.Z., S.W., D.W., S.B., L.B., and E.P.W. cloned, expressed, and purified the proteins. M.Y. carried out all structural and biochemical studies. M.Y., A.C.W.P., and D.B.S. analyzed the structures. Z.L. carried out all optical microscopy experiments. M.Y. and D.B.S. wrote the manuscript. All authors participated in manuscript discussion and editing. J.M.E., B.M.K., D.I.S., and D.B.S. supervised the research.

## Competing interests

The authors have no competing interests
