## [Transparent Peer Review file · Nature Communications]

Amino acid and viral binding by the high-affinity Cationic Amino acid Transporter 1 (CAT1) from *Mus musculus*

Corresponding Author: Dr David Sauer

Version 0:

Reviewer comments:

Reviewer #1

(Remarks to the Author)

This work reports the cryo-EM structures of MmCAT1 in complex with the receptor binding domain from the Friend Murine Leukemia Virus or the substrates, or in the apo state. This study checks the binding affinity of MmCAT1 to RBD of FrMLV. By extensive structural analysis, the authors clarify interaction residues to determine the virus' rodent-specific tropism. In addition, this study solved the complex structure of MmCAT1/FrMLV RBD under different substrate-binding states or the apo state, revealing the mechanisms of the substrate selectivity of MmCAT1 and the conformational changes between the inward-facing open and the inward-facing occluded state. Overall, this is an excellent work, in which only a few issues should be addressed.

1. There are several different spellings for b0,+AT1, such as b0,+AT or B0,+AT1, which should be consistent.
2. Page 5, "... and we hypothesize that when cholesterol is bound at this location the region occupied by the succinate moiety of CHS would become filled with water." There is no experimental data to support this point. Please remove this sentence.
3. Page 11, "Finally, our results suggest a novel mechanism for CAT1's role in cellular sensing of amino acid depletion. TM4SF5 interacts with CAT1 and SLC38A9 in an arginine dependent manner, shuttling between the plasma membrane and lysosome to signal the cells' arginine status to the mTOR complex. MmCAT1's large substrate dependent conformational change suggests the transporter's arginine binding state may change the stability of the TM4SF5:CAT1 complex. Thereby, CAT1 may act as an arginine transceptor, regulating TM4SF5's translocation to the lysosome to modulate mTOR signaling in response to the amino acid's concentration in the extracellular space or cytoplasm." There is no experimental data to support this point. Please remove these sentences.

Reviewer #2

(Remarks to the Author)

The authors provide new and relevant structural data for SLC7A1 in substrate bound and apo state, using mouse CAT-1 bound to the receptor binding domain of Friend Murine Leukemia Virus. A major point is, that there are no functional data provided to show that mutations of putative substrate binding sites affect the ability of CAT-1 to transport substrate.

Other points:

- To what extent does the binding of the viral receptor binding domain change the transporter properties of SLC7A1?
- Do the data allow to speculate if the transport function, in particular the transport-related conformational changes are necessary for the function as a virus receptor?

Abstract:

- Ornithine is not a proteinogenic amino acid and thus does not influence protein structure.
- Is there really agmatine production in mammalian cells?
- The list of functions of cationic amino acids and their metabolites seems somewhat random. You can very well do without it.
- What is meant by aberrant intake of these amino acids and how is this linked to the conditions described?
- y+LAT1 and 2 export cationic amino acids rather than importing them.
- "Among the sodium-independent CAT transporters" sounds like there were also sodium-dependent ones.
- The complex formation of CAT-1 with eNOS (Ref 14) has never been verified, to my knowledge.

Results:

- Please explain FrMLV-RBD at first mention
- Please add reference for GABA-transporting SLC7A14
- Are the residues suggested to be involved in substrate binding conserved among orthologs and paralogs that transport also cationic amino acids?

Discussion:

- There are no data provided to show that substrate-mediated conformational changes change the binding of CAT-1 to TM4SF5. The postulation of a role of CAT-1 as an arginine transporter is thus highly speculative.
- Glu106 of mouse CAT-1 has been shown to be crucial for transport function [1]. Do the data provided here allow to explain the role of this conserved residue?
- Similarly, conserved cysteine residues (Cys 33 and Cys 273 in CAT-2A) have been identified to mediate NEM-inhibition of this transporter [2]. Do the data provided here allow to explain the role of these conserved residues for transport function?

1. Wang, H., M.P. Kavanaugh, and D. Kabat, A critical site in the cell surface receptor for ecotropic murine retroviruses required for amino acid transport but not for viral reception. *Virology*, 1994. 202(2): p. 1058-60.

2. Beyer, S.R., et al., Identification of cysteine residues in human cationic amino acid transporter hCAT-2A that are targets for inhibition by N-ethylmaleimide. *J Biol Chem*, 2013. 288(42): p. 30411-9.

Version 1:

Reviewer comments:

Reviewer #1

(Remarks to the Author)

The authors have addressed the points raised by this reviewer. And this reviewer has no further comments.

Two reviewers have provided strong support for our study of MmCAT1, its substrate selectivity, and interaction to the receptor binding domain of Friend Murine Leukemia Virus. Their expert comments have greatly enriched the revised manuscript, and we are grateful for their feedback.

Reviewer #1 (Remarks to the Author):

This work reports the cryo-EM structures of MmCAT1 in complex with the receptor binding domain from the Friend Murine Leukemia Virus or the substrates, or in the apo state. This study checks the binding affinity of MmCAT1 to RBD of FrMLV. By extensive structural analysis, the authors clarify interaction residues to determine the virus' rodent-specific tropism. In addition, this study solved the complex structure of MmCAT1/FrMLV RBD under different substrate-binding states or the apo state, revealing the mechanisms of the substrate selectivity of MmCAT1 and the conformational changes between the inward-facing open and the inward-facing occluded state. Overall, this is an excellent work, in which only a few issues should be addressed.

1. There are several different spellings for b⁰,+AT1, such as b⁰,+AT or B⁰,+AT1, which should be consistent.

We thank the reviewer for pointing out the inconsistent spelling of b⁰,+AT1. This has been corrected in the revised manuscript

2. Page 5, "... and we hypothesize that when cholesterol is bound at this location the region occupied by the succinate moiety of CHS would become filled with water." There is no experimental data to support this point. Please remove this sentence.

The reviewer is correct that we have potentially over-reached with this hypothesis. We have removed this statement, and are grateful for their observation.

3. Page 11, "Finally, our results suggest a novel mechanism for CAT1's role in cellular sensing of amino acid depletion. TM4SF5 interacts with CAT1 and SLC38A9 in an arginine dependent manner, shuttling between the plasma membrane and lysosome to signal the cells' arginine status to the mTOR complex. MmCAT1's large substrate dependent conformational change suggests the transporter's arginine binding state may change the stability of the TM4SF5:CAT1 complex. Thereby, CAT1 may act as an arginine transceptor, regulating TM4SF5's translocation to the lysosome to modulate mTOR signaling in response to the amino acid's concentration in the extracellular space or cytoplasm." There is no experimental data to support this point. Please remove these sentences.

We are grateful to the reviewer for pointing out the lack of experimental evidence in our initial proposal of CAT1 as an arginine transceptor. A similar concern was raised by Reviewer #2. This section has been rewritten to avoid over-reaching.

Reviewer #2 (Remarks to the Author):

The authors provide new and relevant structural data for SLC7A1 in substrate bound

and apo state, using mouse CAT-1 bound to the receptor binding domain of Friend Murine Leukemia Virus.

A major point is, that there are no functional data provided to show that mutations of putative substrate binding sites affect the ability of CAT-1 to transport substrate.

The reviewer correctly points out that we did not have any substrate uptake assays in our initial manuscript. We have systematically tested several experimental systems for measuring CAT1's transport activity, including both in vivo over-expression and proteoliposome assays with purified proteins and quantified using radioactive substrates or metabolomics. These took a significant amount of time and resources to test and optimize, and the difficulty of these assays is consistent with previous reports of CAT1's low transport activity (Closs et al. Biochemistry 1997).

Nevertheless, in the revised manuscript we demonstrate in vitro 3H-Lys uptake assay by purified MmCAT1 in proteoliposomes (Figure 1c). To complement this, we have repeated the thermostability experiments for several mutants with Lysine. However, it is not feasible to evaluate uptake of the mutants with this assay given the low activity of MmCAT1, the limited throughput of this method, and the need to express and purify large quantities of each mutant. Accordingly, we have revised the text to clarify that our binding assays cannot be used to determine the effects of mutations on CAT1's transport activity.

Other points:

- To what extent does the binding of the viral receptor binding domain change the transporter properties of SLC7A1?

The reviewer raises a key question about the effect of FrMLV-RBD binding on CAT1's native transport activity. The distant binding site of FrMLV-RBD from the transport domain, and poor conservation of this region, indicate this region of the receptor is not essential for the native import activity of MmCAT1. This is in agreement with previous studies showing MmCAT1's amino acid transport is not affected by binding to whole virus or its envelope glycoprotein (Wang et al. Nature 1991). This point has been added to the revised text, and we are extremely grateful for the reviewer's insightful question.

- Do the data allow to speculate if the transport function, in particular the transport-related conformational changes are necessary for the function as a virus receptor?

FrMLV-RBD's binding to a poorly conserved extracellular loop suggest that this protein-protein interaction is unlikely to be affected by amino acid binding or transport. This is supported by our observation of stable binding of the viral protein in two conformations of MmCAT1, and by previous studies showing a transport-inactive mutant still bound MLV protein (Wang et al. Virology 1994). In contrast, the broad tropism of Bovine Leukemia Virus suggests it binds to the conserved transport domain (Bai et al. FASEB J 2019), a notion supported by the effect of BLV-RBD binding on amino acid transport (Ivanova et al. 2023). These points have been expanded upon in the revised manuscript.

Abstract:

- Ornithine is not a proteinogenic amino acid and thus does not influence protein structure.

We thank the reviewer for pointing out this miswording in our description of ornithine's physiological functions. We have corrected this in the revised manuscript.

- Is there really agmatine production in mammalian cells?

As the production of agmatine from arginine is not essential to the background of CAT1, we have revised the manuscript to remove a mention of this molecule. We thank the reviewer for pointing out the unknown relevance of agmatine to mammalian metabolism.

- The list of functions of cationic amino acids and their metabolites seems somewhat random. You can very well do without it.

We thank the reviewer for pointing out that our abstract and introduction could be improved. We have streamlined and focused these points in the revised manuscript.

- What is meant by aberrant intake of these amino acids and how is this linked to the conditions described?

We thank the reviewer for pointing out this sentence was unclear. As the underlying point is ultimately not central to our study of CAT1, we have removed this sentence in the revised manuscript.

- γ -LAT1 and 2 export cationic amino acids rather than importing them.

We thank the reviewer for pointing out that our original phrasing mistakenly described γ -LAT1 and γ -LAT2 as importers. We have revised the text to clarify that these are cationic amino acid transporters, encompassing both importers and exporters.

- "Among the sodium-independent CAT transporters" sounds like there were also sodium-dependent ones.

The reviewer correctly points out that our initial language implied mechanisms on other amino acid transporters. We have revised the text to clarify this point.

- The complex formation of CAT-1 with eNOS (Ref 14) has never been verified, to my knowledge.

Reviewing the primary literature, we concur that there is a lack of evidence for a direct interaction between eNOS and CAT1. We have revised the text to clarify this point.

Results:

- Please explain FrMLV-RBD at first mention

We thank the reviewer for pointing out that this acronym had not been defined at the first mention. This has been corrected.

- Please add reference for GABA-transporting SLC7A14

We thank the reviewer for pointing out that references for SLC7A14 were missing from the manuscript. This has been revised.

- Are the residues suggested to be involved in substrate binding conserved among orthologs and paralogs that transport also cationic amino acids?

The generality of substrate interactions among cationic amino acid transporters is a key insight of our study, and we thank the reviewer for pointing out that their conservation was unclear in the initial submission. We have modified the text, added Extended Data Fig. 3g and revised the Supplementary Sequence alignment to clarify this point.

Discussion:

- There are no data provided to show that substrate-mediated conformational changes change the binding of CAT-1 to TM4SF5. The postulation of a role of CAT-1 as an arginine transceptor is thus highly speculative.

We are grateful to the reviewer for pointing out the lack of experimental evidence in our initial proposal of CAT1 as an arginine transceptor through an interaction with TM4SF5. A similar concern was raised by Reviewer #1. This section has been rewritten to avoid over-reaching on CAT1's function.

- Glu106 of mouse CAT-1 has been shown to be crucial for transport function [1]. Do the data provided here allow to explain the role of this conserved residue?

The reviewer raises an intriguing question about our structure's ability to predict the mechanistic effect of the glutamate mutation that blocks transport of MmCAT1. Examining our structure, this glutamate is at the cytoplasmic face of the protein and immediately adjacent to the latch peptide. Therefore, mutations of this residue can lead to altered interactions with the latch peptide, thereby affecting the energetics of the transport cycle. We thank the reviewer for highlighting this previous observation, and a discussion of the predicted effects for this mutation is included in the revised manuscript.

- Similarly, conserved cysteine residues (Cys 33 and Cys 273 in CAT-2A) have been identified to mediate NEM-inhibition of this transporter [2]. Do the data provided here allow to explain the role of these conserved residues for transport function?

Indeed, our structure immediately explains how NEM modification of equivalents to Cys33 and Cys273 will arrest transport in the CAT transporters. These residues are located on or near TM1a and TM6b and thereby are critical for access to the binding site and gating of the protein. Accordingly, modifications of these residues would arrest the transport cycle. We have added a discussion of this inhibitory mechanism in the revised manuscript.

Two reviewers have previously provided strong support for our study of MmCAT1, its substrate selectivity, and interaction to the receptor binding domain of Friend Murine Leukemia Virus.

Only one reviewer provided feedback on the revised manuscript, stating they were completely satisfied with our changes.

Reviewer #1 (Remarks to the Author):

The authors have addressed the points raised by this reviewer. And this reviewer has no further comments.

We thank the reviewer their feedback and their approval of our revised manuscript.